# Location and Creation of Nest Sites for Ground-Nesting Bees in Apple Orchards

**DOI:** 10.3390/insects14060490

**Published:** 2023-05-24

**Authors:** Michelle T. Fountain, Konstantinos Tsiolis, Celine X. Silva, Greg Deakin, Michael P. D. Garratt, Rory O’Connor, Claire Carvell, Richard F. Pywell, Michael Edwards, Simon G. Potts

**Affiliations:** 1NIAB, New Road, East Malling, Kent ME19 6BJ, UK; k.tsiolis@pgr.reading.ac.uk (K.T.); celina.silva@niab.com (C.X.S.); greg.deakin@niab.com (G.D.); 2Centre for Agri-Environmental Research, School of Agriculture, Policy and Development, University of Reading, Reading RG6 6EU, UK; m.p.garratt@reading.ac.uk (M.P.D.G.); rorysoc@gmail.com (R.O.); s.g.potts@reading.ac.uk (S.G.P.); 3UK Centre for Ecology & Hydrology, Benson Lane, Wallingford, Oxon OX10 8BB, UK; ccar@ceh.ac.uk (C.C.); rfp@ceh.ac.uk (R.F.P.); 4Lea Side, Carron Lane, Midhurst GU29 9LB, UK; ammophila@macace.net

**Keywords:** *Andrena*, blossom, *Lasioglossum*, Gala, habitat management, insect, pollinator

## Abstract

**Simple Summary:**

Wild, ground-nesting bees are important pollinators of apple. They generally do not fly more than a couple of hundred meters from their nest, yet little is known of their preferred nest locations in apple orchards. This project set out to identify where ground-nesting bees are found in apple orchards, which species are present and what influences where they nest. Most of the bee nests were found in the bare ground underneath the apple trees. This area is normally maintained weed-free using a herbicide or mechanical weeding. In addition, fourteen species of ground-nesting bee were identified in the orchards. Hence, maintaining areas of bare ground in apple orchards during peak nesting times could improve nesting opportunities for ground-nesting bees and potentially improve pollination.

**Abstract:**

Wild ground-nesting bees are key pollinators of apple (*Malus domestica*). We explored, (1) where they choose to nest, (2) what influences site selection and (3) species richness in orchards. Twenty-three orchards were studied over three years; twelve were treated with additional herbicide to increase bare ground with the remainder as untreated controls. Vegetation cover, soil type, soil compaction, nest number and location, and species were recorded. Fourteen species of ground-nesting solitary/eusocial bee were identified. Most nests were in areas free of vegetation and areas treated with additional herbicide were utilised by ground nesting bees within three years of application. Nests were also evenly distributed along the vegetation-free strips underneath the apple trees. This area was an important ground-nesting bee habitat with mean numbers of nests at peak nest activity of 873 per ha (range 44–5705), and 1153 per ha (range 0–4082) in 2018 and 2019, respectively. Increasing and maintaining areas of bare ground in apple orchards during peak nesting events could improve nesting opportunities for some species of ground-nesting bee and, combined with flowers strips, be part of a more sustainable pollinator management approach. The area under the tree row is an important contributor to the ground-nesting bee habitat and should be kept bare during peak nesting.

## 1. Introduction

Apple (*Malus domestica*) is a globally important fruit crop covering 5.5 M ha in 2020, a three-fold increase since 1961 [1]. Production was valued at 47,323 USD/tonne (producer price) in 2020 [1] with production in the UK being worth over GBP 150 M per annum on ~23 K ha of land area [2]. Most apple varieties are self-incompatible [3] and reliant upon cross pollination for improved fruit quality and yield [4,5,6]. Insects are essential pollinators of apple, in the UK, contributing GBP 92.1 M per annum [7]. It was estimated for the variety Gala, that there is potential for insect pollination services to improve UK output by up to GBP 5.7 M per annum [5]. However, the dependence of yield and fruit quality on pollinators varies between apple varieties [8].

Wild bees are major contributors to apple pollination [9], but insect guilds differ in their contribution to this ecosystem service [10] and their economic value to UK apple production, according to recent estimates for solitary bees (GBP 51.4 M), honeybees (GBP 21.4 M), bumblebees (GBP 18.6 M) and hoverflies (GBP 0.7 M) [7].

Although honeybees are often more abundant in apple orchards [10], ground-nesting andrenid bees are important pollinators, being both abundant and effective at transferring pollen between flowers [7,9]. In addition, larger andrenid species are thought to carry more pollen on their bodies compared to honeybees [11]. Although pollen load is not always a good predictor of pollen deposited on stigmas, pollination success is likely due to the amount of stigmal contact the insect makes during a flower visit [12]. Ground-nesting bee species that can be solitary or eusocial (e.g., some halictid species) are often found nesting in bare soils with good sun exposure [13]. In addition, many ground-nesting bee species need to nest near to a floral resource as the distance they can optimally forage is shorter than that of honeybees and bumblebees and is size-dependent [14,15,16].

Apple production is often sub-optimal due to insufficient insect pollination [8], and this might be corrected through orchard management practices such as the addition of floral resources and nesting habitats [17,18]. A recent review by Antonine and Forrest [13] highlighted the breadth of nesting habitat requirements among species, including soil substrates and proximity to floral resources, but many ground-nesting bee species are understudied. Most species spend much of their lifecycle in immature stages below-ground, only emerging for a short period (normally a few weeks) to mate, forage for pollen and nectar to provision nest cells and ultimately lay eggs. The habitat preferences of some species have been identified, but these are usually easily visible ground-nesting aggregations rather than less visible species [19].

To maintain abundant populations of functionally important pollinators in agricultural landscapes, provision of both nesting and floral resources is required. While the establishment of additional floral resources in fruit systems has been relatively well-studied [17,20,21,22], the extent to which orchards and associated habitats support ground-nesting bees is not well known. To support the more sustainable production of apple crops and reduce potential pollination deficits, we need to better understand and potentially promote bees nesting in and around orchards.

The aims of our study were to (1) characterise where ground-nesting bees choose to nest in apple orchards, (2) explore what influences their choice of nest site, (3) identify which bee species are nesting in apple orchards, and (4) explore how we can improve habitat management for ground-nesting bees over time.

## 2. Materials and Methods

### 2.1. Study Design

Twenty-three conventionally managed commercial apple (*Malus domestica*) orchards (cv. Gala apple) in Kent, UK, were used for the study. In the original design, the orchards were assigned to one of four treatments in 2016: (1) wildflower intervention, (2) nesting intervention, (3) both wildflower and nesting intervention, and (4) an untreated control. The untreated orchards received no specific pollinator management intervention [17,23]. There were six replicates of each treatment in the initial design except for treatment 1 where one wildflower sowing failed to establish; this was not included in the study design here. For this study, we divided the orchards into those that had received a nesting intervention (treatments 2 and 3) and those that did not (treatments 1 and 4). Orchards were assigned into six study blocks of four orchards grouped by spatial proximity, resulting in two (pseudo)replicates of each treatment in each block. There were 12 orchards with a nesting intervention and 11 orchards without (size range: 0.47 ha to 6.43 ha (mean = 2.05 ha)). All were managed by the growers using standard conventional management, and had a minimum separation distance of 300 m between the edges of study plots within blocks and at least 1 km of separation between blocks to prevent the interactions of bee nesting between study plots [14,24].

During ground-nesting bee assessments, orchards were sampled together in their study block, and the order of visits to each orchard within each study block was randomised for each survey round. Two orchards used in 2017 (one control and one nesting intervention) were unavailable in 2018 and were replaced by different orchards in the same block meeting our study criteria.

### 2.2. Nesting Habitat Treatment

Within each orchard there was a study plot of 40 m × 50 m along one edge of the tree rows, from which the assessments were conducted (Figure 1). The nesting intervention treatment increased the availability of bare-ground nest sites for ground-nesting bees, by extending the herbicide spray strip under the orchard tree rows beyond the row for the full length of each treated orchard. Glyphosate (5l ha^−1^) was applied to the ground in spring 2017 beyond the last tree in the row, resulting in 2.2–3.3 m^2^ of additional bare ground per tree row. Excess vegetation was removed from the nesting plots by raking the surface, so that areas were 80% clear of vegetation from before apple blossom (April) to the middle of the summer (July). The herbicide treatment was re-applied before blossom each year [23].

### 2.3. Nest Density Assessments

Five orchard habitat types were surveyed: those (1) under trees in the herbicide strip, (2) vegetated alleyways, (3) the existing herbicide area at the end of the tree row (every other row), (4) the extended area (every other row), and (5) vegetated headlands (Figure 1). The main vegetation in the headlands and alleyways was mown grass sward. In 2017, numbers of nests were assessed. Six replicate samples of 50 cm × 50 cm (divided into 5 cm squares) quadrats in or adjacent to the study plot were collected in each orchard habitat type (NB: the headland areas were outside the study plots).

In 2018 and 2019, the sampling strategy was changed from quadrats, which was time-consuming and resulted in many zero values, to active searches standardised by spatial coverage in defined areas. Based on knowledge from 2017, we omitted headlands (type 5) and focused on areas in the study plots (1) under trees in the herbicide strip, (2) the existing herbicide area at the end of the tree row, (3) the extended herbicide area, and (4) the unextended areas in the control orchards (typically grassed headland) (Figure 1).

To assess the numbers of bee nests in the tree row area, two people walked slowly on either side of the tree row from the row edge to the end of the study plot, covering 50 m, recording all ground-nesting bee holes. Bee nests were recognised by their tumuli or when closer inspection confirmed they were not made by other fauna (e.g., earthworms and ants). Two rows per study plot were assessed in 2018 and one row per plot was assessed in 2019 as data collected from one row were adequate. The distance from the (a) tree trunk to the margin of the alleyway, and (b) orchard edge, for each nest, was recorded to the nearest cm to ascertain where in the tree row ground-nesting bees preferred to nest.

Nest density was assessed in every existing (yellow area, Figure 1) area and in extended and equivalent but vegetated unextended areas (brown area, Figure 1) of each study plot. The whole area at the end of the 8 tree rows of the study plot was measured (length and width) and searched for bee nests on each sampling occasion. In the equivalent vegetated unextended area, a closer search, at ground level, sorting through vegetation with fingertips, was needed as nests were not easily visible.

Nesting assessments were performed during three apple tree phenological periods: 1. pre-apple blossom (before petals were visible on the developing flower trusses), 2. during blossom, and 3. post-blossom (when the petals had withered and fallen). One to three survey rounds took place for each tree phenological period, each year, depending on the length of phenological period (Table 1). Nest density assessments were occasionally restricted by weather conditions.

### 2.4. Bee Identification

To give an indication of species nesting in each habitat, in 2018 and 2019, the internal diameter of the nest entrance/exit hole was measured to the nearest 0.5 mm [25]. In the final year of the study, we deployed bee traps in each habitat type. Each orchard was surveyed on the same dates as nest surveys were carried out, hence they were surveyed on six occasions (Table 1). At least 50 bee traps were deployed over nest entrances, per visit, depending on nest availability and were constructed from 7 cm wide and 4 cm tall conical mesh strainers (supplied by NISBETS, https://www.nisbets.co.uk (accessed on 30 January 2018)). A 7 mm hole was made at the top and a 50 mL polypropylene multi-purpose container (55 mm × 44 mm, Sarstedt) was secured over the top with a rubber band. Bees emerging from nests flew up into the container where they could be observed. Traps were checked after ~30 min and bees were identified to the species level using the keys by Else and Edwards [26]. *Lasioglossum calceatum* and *L. alpipes* were grouped together [27]. Uncertain identifications were checked by expert taxonomist Michael Edwards.

### 2.5. Soil and Vegetation 

Soil texture (particle size distribution) was measured by sampling a homogenised subsample (10 g) from 5 soil cores, taken from the study area in each orchard, at a maximum depth of 20 cm. Samples were analysed, by laser granulometry (range 0.01–2000 microns, Malven Mastersizer 3000). Samples were analysed at the University of Reading from each of the 23 orchards [28]. Orchard soils in this study were either sandy-silt-loam or silt-loam soils.

Soil compaction in the five habitats in each orchard was measured in 2017, using a handheld penetrometer (Solutions for Research Ltd., Bedfordshire, UK) set to 20 cm. Only measurements that were successful up to 20 cm depth were analysed to reflect the typical nesting depth of our target ground-nesting species [29]. Six replicate samples per habitat per orchard were collected.

To estimate ground cover (bare earth, green vegetation, moss and thatch (dead vegetation)), in the first year, 6 replicate quadrats (50 cm × 50 cm) for each of the five habitat types (Figure 1) were taken. In 2018 and 2019, there were 8 replicate quadrats for each of the 3 habitat types at the ends of the rows (existing, extended, and unextended), on each survey round, with 3 and 4 quadrats under trees per site, respectively.

### 2.6. Statistics

Most of the data had count values as the dependent variable and therefore statistical models appropriate for count data were used. For all count data models, a Poisson distribution was first fitted. Then, if the model was overdispersed, which was assumed when the ratio of sum of squares of the Pearson residuals to residual df was >2, the model was refitted using the negative binomial distribution, as indicated in the text.

To estimate the effect of orchard habitat on bee nesting density, a Poisson generalized linear mixed model (GLMM) was fitted to the habitat data using the glmmTMB package [30] in R (4.1.1, R Core Team 2021). The 2017 data were analysed separately from those of the subsequent years due to the different sampling strategies and the exclusion of one of the habitats in the latter. The dependent variable was the total number of nests; to estimate nest density, it was calculated as the number of nests per m^2^, and an offset equalling each sample size area was added to the model. The models included fixed effects for block, habitat, the 2018/2019 year and the year–habitat interaction in order to test for the differential effects of the habitats on nest density over time. Both models included random effects for orchard. The design used repeated measures, so a further random effect of the interaction between orchard and habitat was used. Finally, there was pseudo-replication within habitats; therefore, a final random effect was used to account for the non-independence of the samples taken within each block, at each assessment point for each habitat type.

To estimate the overall effect of the additional herbicide treatment (extended area) on nest density, a negative binomial (linear mean–variance relationship) GLMM was fitted to data from all years using glmmTMB. The model included fixed effects for year, treatment and the year–treatment interaction. Random effects were included for orchard and the interaction between block, treatment and assessment date, to account for repeated measures and pseudo-replication within each block, respectively.

The relationship between nest density and soil compaction was explored through a linear mixed effect model (LME4 package [31]). The mean nest density per m^2^ of each orchard habitat was calculated, and so a square root transformation was applied. Nest density was then used as the response variable. Soil compaction and surveyed orchard habitats including interactions were included as predictor variables, and orchards were included as a random factor. Dunn’s test of multiple comparisons (rstatix package [32]) tested whether or not there was a significant difference in soil compaction between orchard habitats.

Vegetation cover was estimated as the percentage cover equalling 100. Data were analysed using principle components analysis (PCA), where principal component 1 and principal component 2 were included in the GLMM as predictor factors. Nest density was calculated with the total number of nests as the response factor and the size (m^2^) of each surveyed area as an offset. Orchards and surveyed areas were set as random factors. The dominant habitat type was determined by choosing the habitat with the highest percentage vegetation cover from each assessment. Then, the mean nest density of each dominant habitat was calculated. A Kruskal–Wallis rank test was used to determine whether or not the dominant habitats differed in nest density, and Dunn’s test of multiple comparisons (rstatix package [32]) was used to identify the preferred nesting habitats.

To establish if bees were equally distributed along the tree row, numbers of nests were grouped per 5 m from the start of the tree rows (0 m) and tested between distances using a Chi-square goodness of fit test; post hoc tests were performed between the first group (0–5 m) and each other group with *p*-values adjusted for multiple testing using the Bonferroni method. Equality of distance from the tree and grass alleyway of nest distribution was tested with a binomial exact test. To compare the bee nest entrance/exit diameters between orchards habitats, a linear mixed model was fitted to the data using the LME4 package. The model included fixed effects for block, year, tree–stage and habitat, and the interactions for the latter included three fixed effects. The random effects were orchard, orchard:habitat and block:tree–stage:habitat:assessment to account for orchard random effects, repeated measures random effects and pseudo-replication, respectively. Finally, we plotted the numbers of individual bees captured emerging from nests in the different habitat types and the mean nest hole size from each emerging bee species.

To estimate the nest numbers (response variable) at tree stages (apple blossom timing), a Poisson GLMM was fitted to the data using the glmmTMB package in R [30]. To calculate the mean numbers of nests at peak nesting (sampling time, when most nests were present) under the tree row, the total area of bare earth (under tree row) per orchard was calculated from the row length and width multiplied by the numbers of rows per orchard from ground measurements and Google Maps, respectively. The maximum number of nests per m^2^ was then multiplied by the mean area of bare ground per orchard to give the estimated maximum number of ground-nesting bee nests per ha^−1^ of orchard. The model included fixed effects for block, year, tree–stage and the year:tree–stage interaction, and random effects were included for orchard and block:tree–stage:assessment date to account for orchard (and repeated measures) random effects and pseudo-replication within each block, respectively.

For all GLMMs, the statistical significance of the fixed effects was tested using a likelihood ratio test. For all analyses, post hoc testing was carried out using the R emmeans package [33] with *p* values adjusted for the familywise error rate using the Tukey method.

## 3. Results

### 3.1. Where and When Do Ground-Nesting Bees Nest?

In the first year, there was a significant effect of habitat type on nest density (Chisq 28.37, *p* < 0.001, Figure 2). More nests were observed in the herbicide-treated existing area at the end of the tree row and under the trees, than in the vegetated orchard headlands (z 3.857, *p* < 0.001, Figure 2). Although more nests were observed in the existing area compared to the extended area at the end of the tree row, this was not quite significant (z 2.714, *p* = 0.052, Figure 2).

In 2018 and 2019, there was a significant effect of habitat type on nest density (Chisq 37.60, 3, *p* < 0.001), but there was also an interaction between year and habitat type (Chisq 14.07, 2, *p* < 0.001, Figure 2), likely due to the relatively fewer nests under trees in 2019 compared to 2018 (z 2.875, *p* = 0.070). In general, more bee nests were observed in herbicide-treated existing, extended, and under-tree areas than in areas not treated with herbicide (unextended) (2018: z 5.194, *p* < 0.001; z 3.510, *p* = 0.003; z 4.659, *p* < 0.001, 2019: z 4.851, *p* < 0.001; z 4.474, *p* < 0.001; z 2.801, *p* = 0.026, respectively, Figure 2). In addition, in 2019, there were more nests per m^2^ in the existing and extended areas than under trees (z 3.055, *p* < 0.012; z 2.772, *p* < 0.029, Figure 2).

The density of nests in 2018 between 0 m and 50 m along the length of the herbicide-sprayed area of the apple tree rows did not change significantly. However, there were fewer nests located between 0 and 5 m from the orchard edge, compared to other distances (Chi-sq > 9.3899, 9, *p* < 0.020, Figure 3a). The numbers of nests between the apple tree trunk and the margin of the vegetated alleyway peaked closer to the alleyway than to the tree (Exact binomial test: *p* < 0.001). In total, 57.4% of nests were found to be closer to the alleyways (Figure 3b).

The numbers of active bee nests in the orchards varied significantly between the years and stage of apple blossom (Chisq; year: 19.266, 1, *p* < 0.001; blossom: 21.787, 3, *p* < 0.001). There was also an interaction between year and blossom (Chisq: 34.926, 4, *p* < 0.001). Overall, ground-nesting bees emerged and began nesting pre-apple blossom, peaking during blossom with most nesting activity being completed post-apple blossom (Figure 5a). Based on our nest counts under the tree rows, across all orchards, at peak nesting in 2018 and 2019 there was 873 (range 44–5705), and 1153 (range 0–4082) ha^−^^1^, respectively.

### 3.2. What Influences the Choice of Nest Site?

Soil compaction (kg cm^2^ to 15 cm depth (±SE)) was significantly higher in the headlands and extended herbicide areas at the end of the tree rows compared to the alleyways, under trees or at the end of tree rows (*p* < 0.001, Figure 4a). However, soil compaction did not predict nest density in the apple orchards (LMM fit by REML, *p* = 0.955).

Ground vegetation cover had a significant influence on bee nesting (GLMM, z 14.693, *p* < 0.001). Dominant habitats differed significantly in nest density (Kruskal–Wallis rank test; Chisq 41.522, df = 3, *p* < 0.000). Moss was the dominant ground cover type with the highest nest density, followed by bare earth, thatch (dead vegetation) and then green vegetation (Figure 4b).

### 3.3. Which Bee Species Are Nesting in Apple Orchards?

During 2018 and 2019, a total of 2,289 nest entrances were measured (Figure 5b). Although there was a significant effect of year (F 1.662, *p* < 0.001), blossom stage of trees (F 56.656, *p* < 0.001) and habitat type (F 32.177, *p* < 0.001) on bee nest entrance hole size, there was also a significant interaction between year–blossom stage (F 4.306, *p* = 0.015) and year–habitat (F 8.638, *p* < 0.001). Overall, nests in the unextended area were smaller than nests in the bare ground existing areas (z 5.187, *p* < 0.001), extended areas (z 3.086, *p* = 0.011) and under the trees (z 7.939, *p* < 0.001). Nests under trees were also larger than those in the existing (z −4.127, *p* < 0.001) and extended (z −5.573, *p* < 0.001, Figure 5b) areas.

In the final year of the study, 58 bees from across all sites were captured while leaving nests and identified to the species level. The species identified during the survey in 2019 were *Lasioglossum malachurum* (15), *Andrena flavipes* (10), *Lasioglossum calceatum* (8), *Andrena nitida* and *Andrena haemorrhoa* (5 each), *Andrena dorsata*, *Lasioglossum alpipes*, *Sphecodes monilicomis* (3 each), *Andrena chrysosceles*, *Andrena cineraria*, *Andrena labialis*, *Andrena nigroaenea*, *Andrena trimmerana* and *Lasioglossum fulvicorne* (1 each). Most species were found in the existing bare-ground area at the end of the tree row with *A. nitida*, *A. haemorrhoa*, *L. malachurum*, *L. calceatum*, and *S. monilicomis* being sampled from within the tree row (Figure 6a). This data suggested that the newly created extended bare-earth areas were mostly utilised by the smaller *Lasioglossum* species (Figure 6a,b).

### 3.4. How We Might Improve Habitat Management for Ground-Nesting Bees

The main treatments applied to the orchards were herbicide (extended) vs. no herbicide (unextended) areas at the end of the tree row. Overall, although the numbers of nest entrances were greater in the extended than unextended areas from the second year of the study (Figure 2), there was no significant overall increase in nest numbers (m^2^, ±SE) across orchards that had additional herbicide treatment in any year (no additional herbicide: 0.183 ± 0.043; treated: 0.191 ± 0.045) (Chisq 0.014, 1, *p* = 0.902), even though nest density had increased compared to that in the same area of the unextended habitat by the second and third years (Figure 2B,C).

## 4. Discussion

This study aimed to increase our understanding of where important insect pollinators of apple, namely andrenid bees, nest in apple orchards, to identify approaches to increase bee nest numbers to enable better the targeting of habitat management for apple pollinators within orchard crops.

### 4.1. Where Do Ground-Nesting Bees Choose to Nest and What Influences the Species’ Choice of Nest Site?

More ground-nesting bee nests were observed in the bare-ground areas of orchards, especially the herbicide-treated section under and at the ends of the tree rows. Some ground-nesting bee species utilise bare ground as the soil is warmer than that in vegetated areas, presumably enabling faster development and increased foraging activity (e.g., [34,35,36]). Very few bee nests were observed in the vegetated areas, including the orchard alleyways and headlands. Vegetation may impede digging because of roots in the soil or create a cooler microclimate not suited to bees requiring faster development, especially species that are bivoltine (e.g., [37,38]). We did not search beyond the main cropping area for nests, and future investigations might explore adjacent habitats and perimeter features such as walls, tracks, ditches, and hedgerows. Bare areas and areas of moss, predominantly where herbicide had been applied over many years, were the habitats most likely to host ground-nesting bees.

Peak nesting activity coincided with apple blossom, which is the peak period of nectar and pollen production in the orchards. The ground beneath the trees, at this time, would be unshaded by foliage in contrast to later tree canopy development. Nesting activity generally decreased post-apple blossom. Future studies should include measurements of light levels during the nesting phase to ascertain if light and subsequent soil temperature drive nest site selection in apple orchards. In addition to seasonal changes in nesting, we also observed interannual differences in nest numbers. For example, there were almost half the number of nests under trees in 2019 compared to 2018, over all sampling occasions. The reason for this decline is not known, although rain events were observed to destroy tumuli and may have obscured bee nests. In addition, the stochasticity of bee populations driven by weather, the previous years’ floral resources, parasites and cuckoos cannot be ruled out.

In our study, soil was a poor predictor of nest density. The soils in our 23 study orchards were very similar, sandy-silt-loam or silt-loam soils, and the bee species we encountered are all reasonably common, hence soil type was not a good predictor of nesting in this system. Additionally, soil compaction, in our study, ranged between ~25–50 kg cm^2^ and did not predict nest density in the orchards, unlike in other studies [34,39,40]. Therefore, we suggest that other indicators are better predictors of nest density.

Fourteen ground-nesting bee species were identified using nests during our surveys, three of which were previously recorded as common visitors of the apple blossom [18,41,42,43]. Most species were found in the existing bare-ground area at the end of the tree row and almost half of the individuals caught and identified were from this area. However, due to the area of bare ground under the apple trees, most nests were observed in this habitat type. *Andrena nitida*, *A. haemorrhoa*, *Lasioglossum malachurum*, *L. albipes*/*calceatum*, and *Sphecodes monilicomis* were the dominant species in this habitat. There was little variation in the numbers of nests along the tree row, but bees tended to nest approximately halfway between the tree trunk and the edge of the vegetated alleyway. This might be due to a range of factors including disturbance from farm activities, soil moisture and plant roots. The tree row forms a large part of the orchard surface area and our estimates of ground-nesting bee nests utilising this area, across all orchards, at peak nesting in 2018 and 2019 were 873 (range 44–5705), and 1153 (range 0–4082) ha^−^^1^, respectively. These counts are likely to have been underestimated due to a proportion of the nests being closed or due to disturbance by weather events at the time of assessment.

Data collated by Garratt et al. [7] from UK apple orchards recorded 631 honeybees, 823 solitary bees, 243 bumblebees, 76 hoverflies and 142 others, mostly Diptera. Andrenid bees were the main wild bees visiting apples and accounted for up to two thirds of all visitors. The most abundant ground-nesting species we trapped for identification were smaller species, e.g., *L. malachurum* and *L. calceatum*, and *A. flavipes*, and are probably not as important for apple pollination.

Using nest entrance size as a proxy for bee size, we observed smaller nest entrances in newly created vegetation-free extended areas compared to under the trees. We hypothesise that larger bee species nest under apple trees which would be shaded for most of the apple growing season, with smaller species nesting in open ground. Caution should be observed when interpreting this data as other factors including temperature may play a role in nest entrance size. *Lasioglossum malachurum* exploited our newly herbicide-extended areas at the ends of the tree rows. This species is polylectic and obligately eusocial, and nesting aggregations are common on compacted bare soil [44]. *Andrena flavipes* was present in the more established existing herbicide areas at the ends of the tree row. This species is a bivoltine, so females will be active from March continuously through to August [37,38], hence more exposed warmer soils might aid the development of two generations within a year. *Andrena flavipes* also nests in aggregations in ground exposed to the Sun [37]. *Lasioglossum calceatum* can nest in short turf or open situations [26]. *Sphecodes monilicomis*, also identified in our surveys, is a kleptoparasite of *Halictus* and *Lasioglossum* species [26].

Overall, our findings suggest that bare ground created by herbicide application provides suitable nesting locations for multiple ground-nesting bee species in orchards (unlike the farmland study by [45]), and that the bee species most likely to provide a beneficial ecosystem service through apple pollination chose well-established bare ground under the trees.

### 4.2. How Might We Improve Habitat Management for Ground Nesting Bees?

Applying herbicide to the ends of the tree rows did not increase overall nest numbers in the orchards during the timespan of the study, even though the numbers of nests had increased in the herbicide-extended habitat compared to the same area of the unextended habitat by the second year (Figure 2B). This may be because the extended areas were not large enough compared to the nesting area hosted under the trees and because there was interannual variation in the numbers of nest counts, which were at least partially impacted by weather conditions, or because nest density had already reached maximum.

In addition, these areas extended into the headland, where there is regular vehicular movement. Although newly created areas of bare earth may stimulate nest initiation, traffic can have a negative impact on nest success [34]. Gregory and Wright [46] created four 3 m × 5 m open ground areas in a nature reserve which attracted a few nesting bee species. However, as seen in our study, it is difficult to predict whether or not this augmentation is increasing the abundance and diversity of bees across an area or potentially pulling bees away from other, less favourable nesting locations [35].

Our study highlights the importance of the bare-ground habitat under trees and suggests that this area should be maintained as largely vegetation-free areas, especially during peak nesting times. Nichols et al. [45] compared creating bare-earth sites for bees by either spraying sites with herbicide or scraping the surface vegetation with machinery. Sprayed plots reached over 90% vegetation cover by July compared to scraped plots which reached only 50%. However, no attempt was made to remove the dead vegetation or apply additional herbicide to the sprayed plots. Hence, significantly more nests were found in plots created by surface scraping. Any form of bare-earth nesting habitat will need regular maintenance to keep the ground clear, particularly during the main nesting activity periods. The potential loss of approval of the herbicide glyphosate as a treatment to reduce competition from vegetation with the orchard trees is driving a move towards mechanical weeding. Many machines disturb only the top few centimetres of soil, but studies are needed to assess whether or not mechanical weeding impedes the emergence of overwintered solitary bees in orchard soils.

A review of the impacts of pesticide exposure on solitary bees [47] suggested that glyphosate can affect conditional learning and navigation, as seen in honeybees [48,49]. Most glyphosate studies test honeybees, especially adult bees, through direct contact, or oral dosing experiments [50]. To date, there is one study to suggest that glyphosate may be detrimental to the reproductive success of solitary bees [51]. A recent review by Christmann [52] highlighted that 70% of wild bees nest below-ground and are at risk from chemicals, soil compaction and deep tillage. Deep tillage, reaching 15–30 cm deep, can destroy nests and prevent around 50% of the emergence of ground-nesting bees [53,54]. In combination with studies on the impact of mechanical weeding, it would be helpful to measure the impacts of herbicide on the ground-dwelling stage of bees so that recommendations can be made on the timing and numbers of applications that would be acceptable. Practices such as burning vegetation do not kill most ground-nesting bees [55], and it may be possible in the absence of glyphosate to use herbicides which desiccate the foliage above ground. In addition, new electrical weeding strategies should also be tested for their impacts on ground-nesting bees.

Creation and maintenance of bare-earth areas in apple orchards need to be carried out with the other needs of bee species in mind, for example floral resources [17,56,57,58,59]. There is the potential to manipulate bee populations [60] and target species best-suited to a particular crop by fostering and provisioning the preferred nesting sites and forage of those species. However, in large-scale apple production, nesting habitat, pollen and nectar sources are needed within orchards as solitary bees, overall, do not forage far from the nest. For example, fifty percent of female andrenids, released from a single point, were recovered at distances under 100 m [14]. Smaller solitary bee species probably forage slightly shorter distances from 90 m, but a smaller proportion of bees can be found at 300 m or more away from nesting areas [15]. In addition, pesticides harmful to foraging adults should be avoided where possible at key flight times and care should be taken to avoid tank mixes that exacerbate toxicity [50]. Application of systemic pesticides such as neonicotinoids applied directly to soil should be avoided [61,62].

Nest site selection remains an understudied area of research, but progress has been made with specific species suggesting that softer soils are utilised when there are lower densities of bees, but that larger aggregations need more compacted soils, presumably to maintain structural integrity [34,35].

## 5. Conclusions

In summary, this study has highlighted the importance of bare ground in orchards as a key nesting habitat for bees that are important to apple production. Future studies should focus on designing orchard landscapes that complement the whole lifecycle of bees needed to cross-pollinate apple, and test future changes in orchard practices to ensure that new orchard management approaches do not adversely harm ground-nesting bees.

## Figures and Tables

**Figure 1 insects-14-00490-f001:**
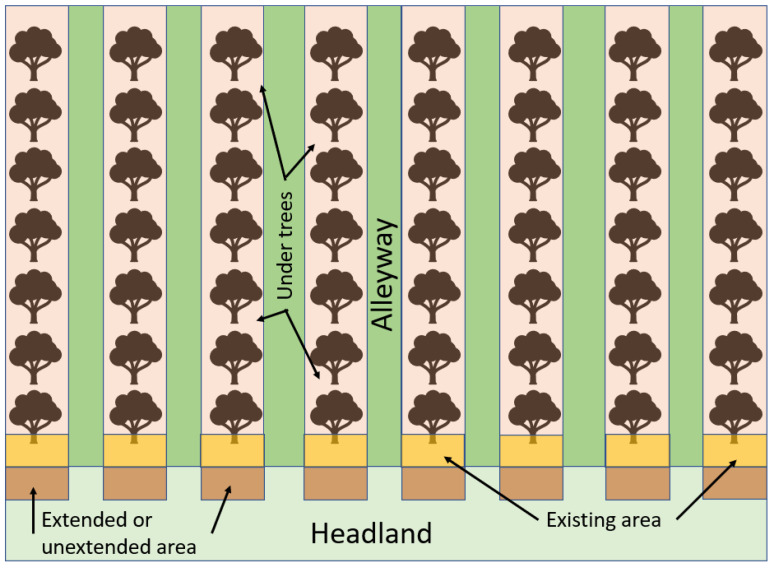
Schematic of a single study plot (40 m across the rows and 50 m along the length of the rows from the row edge) with 5 orchard habitat types assessed in the orchards: (1) under trees in the herbicide strip, (2) vegetated alleyway, (3) existing herbicide area at the end of the tree row (every other row), (4) extended or unextended area, and (5) vegetated headland. In control orchards, the extended herbicide areas (shown in brown) were unextended and hence vegetated. Habitat types 1–5 were assessed in 2017 and 1–4 were assessed in 2018 and 2019.

**Figure 2 insects-14-00490-f002:**
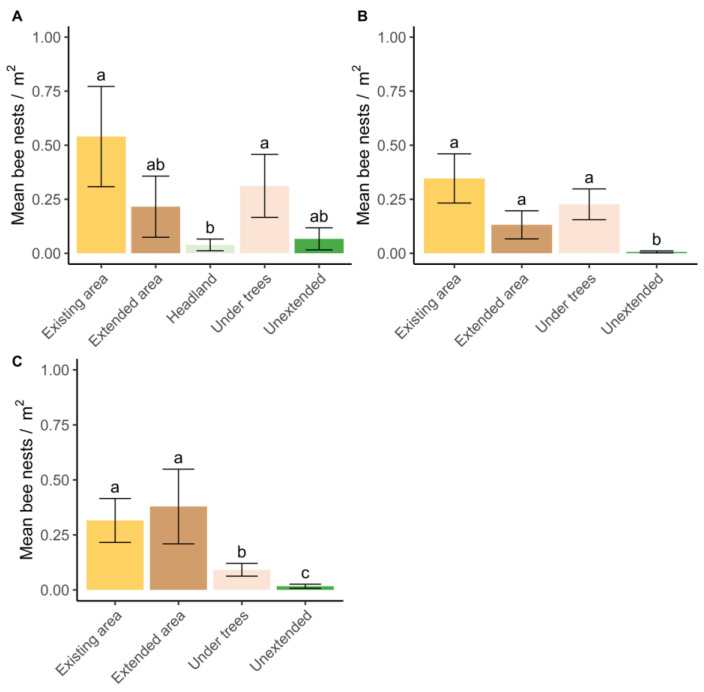
Predicted mean (±SE) density of bee nests (m^2^) in apple orchard habitats (*N* = 23) in (**A**) 2017, (**B**) 2018 and (**C**) 2019. Existing and under-tree areas were treated with herbicide for the life of the orchards and the extended area was treated with herbicide from 2016 (see Figure 1). The headland and unextended areas were vegetated and not treated with herbicide during the study. Nest density was assessed with quadrats in 2017 and then with timed demarked searches in subsequent years. The headland was only assessed in the first year of the study. Bars with different letters (a, b, c) are significantly different.

**Figure 3 insects-14-00490-f003:**
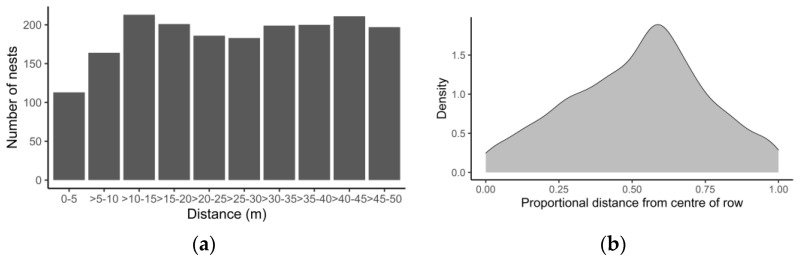
(**a**) Nest density in 2018 in 5 m intervals along the herbicide area of the tree rows for all orchards from the orchard edge (0 m) to the end of the study plot (50 m), and (**b**) density plot of the proportion of bee nests observed between the tree trunk (centre of row, 0.0) and the edge of the vegetated alleyway (1.0), across all orchards.

**Figure 4 insects-14-00490-f004:**
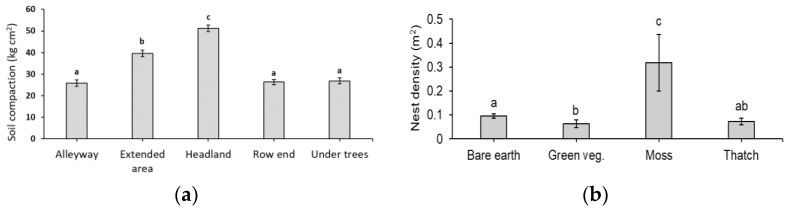
(**a**) Mean (±SE, *n* = 6) soil compaction (kg cm^2^ to 15 cm depth) of orchards’ habitats in 2017. (**b**) Mean (±SE, *n* = 6) bee nest density (m^2^) in areas of bare ground, green vegetation, moss or thatch (dead vegetation). Bars with different letters (a, b, c) are significantly different.

**Figure 5 insects-14-00490-f005:**
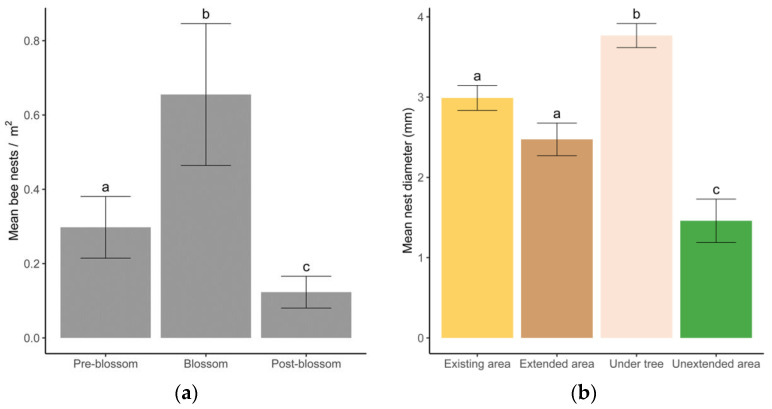
(**a**) Predicted mean (±SE) nest density (m^2^) of ground-nesting bees in orchards in the spring between 2017 and 2019 categorised by the apple tree blossom stage (pre-blossom—no flowers open, blossom—at least 50% flowers open, and post blossom—petals withered). (**b**) Predicted diameter (mm (±SE)) of ground-nesting bee nest entrance holes in the 4 orchard habitats; existing area at the end of the tree rows and under trees treated with herbicide for the life of the orchards. Extended area treated with herbicide from 2016 and the unextended vegetated areas (not treated with herbicide during the study) are provided for comparison (see Figure 1).

**Figure 6 insects-14-00490-f006:**
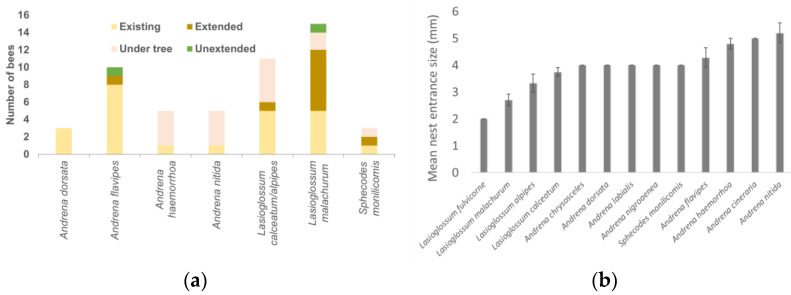
(**a**) Total numbers of bee individuals depending on species and nesting location (excluding where only one was identified) in 2019. Existing area at the end of the tree rows and under trees was treated with herbicide for the life of the orchards. The extended area was treated with herbicide from 2016 and the unextended vegetated areas were not treated with herbicide during the study for comparison. (**b**) Mean entrance size (mm (±SE)) of nest measured to the nearest mm of bees identified to the species level in 2019. Bars with no error bars only had one replicate. NB: *Sphecodes monilicornis* is a cleptoparaitic species and so this is likely the nest diameter of the host.

**Table 1 insects-14-00490-t001:** Dates of nest survey rounds for ground-nesting bees. Note there were only 5 rounds in 2018 due to a shorter nesting period.

Tree Stage	2017	2018	2019
Pre-blossom	4–7 April	17–20 April	29 March–5 April
	10–11 April	23–25 April	8–11 April, 15–18 April
Blossom	19–20 April	1–4 May	23–26 April
	25–26 April	8–11 May	26 April–2 May
Post-blossom	16–24 May, 25–31 May	15–18 May	13–15 May

## Data Availability

All datasets are available upon request from Michelle Fountain and Greg Deakin, NIAB.

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
