# Peer review of "Location and Creation of Nest Sites for Ground-Nesting Bees in Apple Orchards"

_insects, 2023, doi:10.3390/insects14060490_

Round 1

Reviewer 1 Report

The manuscript presents a study of the nesting site locations of ground-nesting bees within United Kingdom apple orchards, including experimental treatment of nesting habitats to explore whether pollinator nesting aggregations can be augmented in situ.  The reporting of results can be improved with further consideration of bee biology and potential exploration of alternate measures of variability in figures.  The manuscript is generally well written and if word-counts/page-numbers are an editorial issue, then document could be considerably streamlined by revising the very detailed Methods section.  Otherwise, this paper provides a refreshing perspective on a usually overlooked, but ecologically important, aspect of pollination service which the authors should be commended for tackling.  

Despite the fact that experimental outcomes were not conclusive, publication of this work should encourage further studies on pollinator biology within agricultural landscapes.  This has major impacts on human food production globally, which the authors emphasize in their Introductory section.  

The following specific comments are provided in good faith, in the hope that they assist the authors to constructively improve the manuscript.

TITLE
Line 2.  The authors do not explore aspects of bee nesting biology that would determine the presence or absence of group nesting, nestmate interaction, cooperative brood care or reproductive skew within colonies of these ground nesting bees.  Hence, whether bees are solitary or social (indeed the authors indicate some study species are bivoltine and/or eusocial) is irrelevant to the aims and outcomes of this research – in this light, suggest removing the word ‘solitary’ from the title and elsewhere in the manuscript.

Suggest reading Michener (1974, 2007), Schwarz et al. (2007), Field et al. (2012) and Danforth et al. (2019) for broad definitions and examples of sociality beyond the corbiculate bee clades.

Michener CD. 1974. The social behavior of the bees: a comparative study. Harvard University Press: Cambridge MA.
Michener CD. 2007. The bees of the world. Johns Hopkins University Press: Baltimore MD.
Schwarz MP, Richards MH, Danforth BN. 2007. Changing paradigms in insect social evolution: insights from halictine and allodapine bees. Annual Review of Entomology 52: 127-150.
Field J, Paxton R, Soro A. et al. 2012. Body size, demography and foraging in a socially plastic sweat bee: a common garden experiment. Behavioral Ecology & Sociobiology 66: 743–756.
Danforth BN, Minckley RL, Neff JL. 2019. The solitary bees: biology, evolution, conservation. Princeton University Press: Princeton NJ.

SIMPLE SUMMARY
Lines 11,14,17,19.  Delete ‘solitary’.

ABSTRACT
Lines 20,25,31.  Delete ‘solitary’.

INTRODUCTION
Lines 58-60.  As far as we know, bee foraging distance has less to do with nesting mode (social/solitary) and more to do with body size related physiology – see Greenleaf et al. (2007) which includes data on andrenids and halictids.  

Greenleaf SS, Williams NM, Winfree R. et al. 2007. Bee foraging ranges and their relationship to body size. Oecologia 153: 589–596.

Line 65.  Solitary bees are not necessarily cryptic, just studied less.  The lack of information is due to a lack of natural history studies generally, social ground nesting bee nests (e.g. halictids) tend to get excavated and studied more because there are deeper philosophical/theoretical research questions being addressed.  

Line 78.  Here and elsewhere, delete ‘solitary’ unless presented in the correct context.

MATERIALS & METHODS
General Comment. This section is very detailed and in some instances verbose, perhaps try to re-write more succinctly especially if page-numbers/word-counts are an issue.

Lines 85-92.  The descriptions of “…the original design…” and “…the initial design…” makes the reader immediately wonder what has changed.  If there are multiple designs, the number of variants should be discussed with associated rationale up-front.

Lines 94-97.  Some repetition of information from first half of the paragraph – consider condensing text.

Lines 102-105.  Not ideal, but beyond the authors’ control.

Lines 109-119.  Suggest making reference to Figure 1 early in this paragraph.  It was difficult to follow the treatment explanation until Figure 1 was referred to on Line-124 and sigh

Lines 122-128.  This is informative, however, the 2017 quadrate methodology is not really comparable to the 2018-19 active search methods.  Consider removing the 2017 data from the manuscript as it does not appear to contribute any additional insight to the overall research outcomes; rather, it seems to just introduce unnecessary noise.  The 2017 data likely required a lot of effort to accrue, however, sometimes less is more.

Line 168-169.  Not sure a table is necessary to convey sampling dates.  This information could just written in the main text body, most readers won’t be concerned if a few days were missed over weekends and so this could be condensed, for example:
“Surveys were undertaken across three years during three bloom phenology stages: pre-blossom (4-11 April 2017, 17-25 April 2018, 29 March -18 April 2019); blossom (19-26 April, etc.)…”

Line 170.  Would ‘Bee identification’ be a more appropriate sub-heading?

Lines 183-184.  Lasioglossum and other halictids are notoriously difficult to distinguish in the field due to conserved morphology and cryptic speciation – see Russo et al. (2015).

Russo L, Park M, Gibbs J, Danforth B. 2015. The challenge of accurately documenting bee species richness in agroecosystems: bee diversity in eastern apple orchards. Ecology & Evolution 5: 3531– 3540.

Line 199.  Superfluous parenthesis.

Line 204.  Were data assessed or visualised for normality (e.g. Kolmogorov-Smirnov or Shapiro-Wilk tests).  Alternatively, if it was assumed a priori that data were non-normal, then perhaps make a brief mention of this.

Lines 256-268.  Not convinced that an effect of apple blossom is a valid assumption for bees founding new nests.  If the study bees were engaged in a strict pollination syndrome (e.g. mono- or oligolectic pollinators) with apple then this might be justified.  However, these bees are very likely to be polylectic and just happen to be situationally nesting near apple (which they visit for a source of proteins and carbohydrates).  Dispersal of female bees from their natal nest and subsequent founding of new nest is more likely to be related to climate, hence, it would be more biologically meaningful to treat this like a natural history life-cycle paper publication would - assess nest founding as a function of physical environmental factors time (month/week) or season or temperature or daylength.  Because these bees have not necessarily evolved over millions of years to emerge with specific reference to human cultivation of apple.  One can verbally associate or correlate the physical factors with the apple orchard bloom phases, but it is unlikely to be a causative effect of nest founding.  A semantic argument, but biologically relevant.

RESULTS
Lines 272-277.  As previously suggested, perhaps delete these results based on the lack of comparative value due to methodological changes from 2017 to 2018-19.  Similarly for Figure 2a.

Lines278-286.  Here and elsewhere, what test do the chi-square values pertain to (Kruskal-Wallis)?
Also, it is not clear what the various z tests are comparing – it might be appropriate to present these results as a table so that comparisons are more easily interpreted.

Line 298.  Shouldn’t this statistic be a series of pair-wise post hoc tests comparing the 0-5m category with all other distance categories?

Lines 307-314.  Further to previous comments about biologically meaningful statistical factors – it is clear from these results that bees are emerging and founding new nests pre-bloom.  This suggests that bee nesting behaviour is not determined by the apple tree flowering phenology.

Line 325.  It is not clear what is being compared with regard to dominant habitats, does this mean there was differences in the number of nests within a plot regardless of the vegetation type.  It’s difficult to interpret the meaning of this result.

Line 326.  What are the degrees of freedom for the Kruskal-Wallis test?

Lines 334 & 336.  Change “nests” to ‘nest entrances’.

DISCUSSION
Lines 375-379.  This statement of aims might be better placed in the Introduction, or at least accompanied by a succinct summary of the main results if kept at the beginning of the discussion.

Lines 394-405.  Disagree with the initial premise, for reasons previously outlined.  Bees were founding new nests in the pre-bloom period, therefore, it is counterintuitive to suggest that new nest founding is “…likely because this was also the peak of nectar and pollen production…”.  Such a conclusion is overreach.
It is, however, entirely reasonable to conclude that bee populations are highly stochastic based on a range of physical and biological factors as the authors state at the end of this paragraph.

Lines 419-421.  Statement that nesting location may be due to soil compaction.  Disagrees with statement on the previous page, namely that “…soil compaction, in our study, …did not predict nest density in the orchards” [lines 409-410].

Lines 433-437.  It is alright to speculate in the Discussion, but there needs to be some further rationale to back it up – otherwise this again seems like overreach.  Can you link the thermoregulation requirements of small bees cf. comparatively larger bees in this context with shade versus non-shade habitats?

Lines 441-443.  Be careful of interpretations here: voltinism relates to the number of brood cohorts reared through in a season and is unrelated to when bees fly.  Egg laying females (solitary or social queens) will lay as many eggs as they can and fly continuously throughout the season to provision brood with mass provisions.  So females will be active from March continuously through to August, populations numbers may increase periodically as newly eclosed females emerge – but I am not sure what the connection is with nest entrance size so perhaps reconsider the argument being presented here.

Lines 467-469.  See comments immediately above.  To get a better handle on voltinism, perhaps read up on its importance to the evolution of eusociality in bees, for example - see Plateaux-Quenu  et al. (1989), Gruber & Field (2022) and references therein.

Plateaux-Quenu C, Plateaux L, Packer L. 1989. Biological notes on Evylaeus villosulus (Hymenoptera, Halictidae), a bivoltine, largely solitary halictine bee. Insectes Sociaux 36: 245–263.
Gruber J, Field J. 2022. Male survivorship and the evolution of eusociality in partially bivoltine sweat bees. PLoS ONE 17: e0276428.

FIGURES
General Comment.  None of the figures have titles.  Consider including a succinct title to prime the reader.

Figure 2a. Suggested deleting.

Figure 3a.  Do the histograms represent absolute values (sums) and were these distance data collected across multiple years?  If so, why not present mean values with indicators of variability (standard errors or 95% confidence intervals).

Figure 3b.  What is the unit of distance on the x-axis (metres)?

Figure 4a.  Perhaps double-check statistical analyses here or use an alternative measures of variability (e.g. 95% confidence intervals) because visual inspection of the figure would suggest that the Extended Area and Headland are significantly different from each other.

Reviewer 2 Report

Your statement that „nest density doubled within areas treated with additional herbicide after three years“ can be misleading for the Abstract readers, as other authors argue, that concurrently „the probability of finding breeding cells was two times higher in nests of solitary bees without glyphosate formulation compared with treated nests“ (Gfaffigna et al., 2020, Apidologie volume 52, pages272–281 (2021) - https://doi.org/10.1007/s13592-020-00816-8). As the above mentioned authors also states , that „glyphosate negatively affects the reproductive success of solitary wild bees“ I also propose remove / reformulate statement on lines 485 – 486 saying, that „to date, there are no data to suggest that applying glyphosate to areas where ground-nesting bees are active has either direct or sublethal detrimental impacts“.

Ideally remove Glyphosate also from Key words. As among funding bodies are stated also producers / distributors of glyphosate formulations, this can help to prevent allegations of conflict of interest.

I am OK with the design and conclusions of this study, the overall feeling is good, but since the study tracks the number of nests, but not the number of individuals fledged from them, I am disturbed by the recommendations for repetitive use of pesticides even in cases where other methods could be equally effective (eg from reference no. 43). If other bee species are affected by glyphosate (f.e. DOI: 10.1126/science.abf748), it is highly expect also to affect solitary bees at some level and it will be worth to do not propose glyphosate application as „good practice“.

Reviewer 3 Report

The authors investigated the importance of turf removal for creating new nesting sites for digger bees in orchards. The results prove that additional plots without vegetation created at the head of tree rows provide new nesting sites for ground-nesting bee species.  The density of nests in these plots is higher than that of vegetation-covered plots. However, treating whole orchards with herbicide to uncover the ground does not cause differences in nest bee density. Nonetheless, the method of protecting bees in the orchard may support the increase in the pollinator abundance of fruit trees because it offers bees new nesting sites outside the tree rows. Enlarging the area of such plots will likely increase the effectiveness of this method. The work indirectly points out the importance of ground-nesting bee species besides the common cavity-nesting bees like mason bees used as managed pollinators.

The manuscript fulfils the formal requirements and is prepared and written carefully but some explanations of the study methods and other issues are necessary.

- In a single plot, were all 8 existing or extended/unextended areas controlled on each date?

If the rows were controlled, there was an alternative choice of other rows on the next date.

However, when the control of all existing and extended areas was repeated after 1-4 days or without an interval (in April 2019), there was a high probability of a repeat count of these nests. Nest construction can take bee female several days depending on the weather.

- How the diameter of the nest entrance was measured? What tools were used to precisely measure cavities in the ground that are easily plugged or if the hole is made fresh there is loose soil around it?

- When were bees collected using ”solitary bee emergence traps”? I guess that it was during the nesting activity when the nest entrance could be found.  However, the name of the trap “solitary bee emergence traps “ suggests that they were trying to catch bees when they first burrowed out of the soil after overwintering. Emergence is the time when adults emerge from their nests, or cocoons after the end of diapause.  I suggest changing the name of these traps omitting the word “emergence”.

Figure 4B. shows new area categories that were not mentioned in the methods. How this relates to the 5 orchard habitat types that were on the study plot (e.g. Fig. 1).  Are bare earth the area under the trees or existing, extended areas? Was nest density determined by the same method?

Line 414. Most species were found in the existing bare-ground area at the end of the tree row.

Also, nearly half of the individuals caught and identified were from the existing area.

Line 415.  However, most bee nests were observed along the herbicide strip under the apple trees.

This interpretation I suggest considering the context of the differences between the compared areas (one row under the trees - 100 m2- my estimate, one existing area 2.2-2.3 m2). Otherwise, the reader may have the feeling that the statement conflicts with the results of the study (Fig. 2.).

- Some more comments are included in the manuscript’s text in the pdf file.
